# Hepatitis C virus seroprevalence, testing, and treatment capacity in public health facilities in Ghana, 2016–2021; A multi-centre cross-sectional study

Yvonne Ayerki Nartey[1,2]*, Rafiq Okine[3], Atsu Seake-Kwawu[4], Georgia Ghartey[5], Yaw Karikari Asamoah[5], Ampem Darko Jnr Siaw[2], Kafui Senya[3], Amoako Duah[6], Alex Owusu-Ofori[7], Opei Adarkwa[8], Seth Agyeman[9], Sally Afua Bampoh[10], Lindsey Hiebert[11], Henry Njuguna[11], Neil Gupta[11], John W. Ward[11], Lewis Rowland Roberts[12], Ansumana Sandy Bockarie[1], Yaw Asante Awuku[13], Dorcas Obiri-Yeboah[14]

1 Department of Internal Medicine, School of Medical Sciences, University of Cape Coast, Cape Coast, Ghana, 2 Department of Internal Medicine, Cape Coast Teaching Hospital, Cape Coast, Ghana, 3 World Health Organisation, Country Office, Accra, Ghana, 4 National Viral Hepatitis Control Program, Ghana Health Service, Accra, Ghana, 5 Ghana Field Epidemiology and Laboratory Training Programme, School of Public Health, University of Ghana, Legon, Accra, Ghana, 6 Department of Internal Medicine, University of Ghana Medical Centre, Accra, Ghana, 7 Department of Clinical Microbiology, School of Medicine and Dentistry, Kwame Nkrumah University of Science and Technology, Kumasi, Ghana, 8 Department of Obstetrics and Gynaecology, Komfo-Anokye Teaching Hospital, Kumasi, Ghana, 9 Department of Biochemistry, Cell and Molecular Biology, University of Ghana, Legon, Accra, Ghana, 10 Department of Internal Medicine, Greater Accra Regional Hospital, Accra, Ghana, 11 Coalition for Global Hepatitis Elimination, Task Force for Global Health, Decatur, GA, United States of America, 12 Division of Gastroenterology and Hepatology, Department of Medicine, Mayo Clinic Rochester, Rochester, MN, United States of America, 13 Department of Medicine, University of Health and Allied Sciences, Ho, Ghana, 14 Department of Microbiology and Immunology, School of Medical Sciences, University of Cape Coast, Cape Coast Ghana

* ynartey@uccsms.edu.gh

**Data Availability Statement:** Data cannot be shared publicly due to confidentiality and local restrictions which prevent sharing abstracted

## Abstract

The current burden of Hepatitis C virus infection and the availability of HCV-related services in Ghana are not well described. Previous estimates on HCV seroprevalence in the country are outdated. This study investigated the HCV seroprevalence and testing and treatment capacity in Ghana. A multi-centre cross-sectional study was conducted in which laboratory and blood bank registers from 17 public healthcare institutions in Ghana were reviewed. A survey on cost and availability of HCV-related testing and treatment was also performed. Crude and pooled estimates of HCV seroprevalence, frequency and median cost of available diagnostic tests and medicines were described. The crude HCV seroprevalence was 2.62% (95% CI 2.53–2.72) and the pooled estimate was 4.58% (95% CI 4.06–5.11) among 103,609 persons tested in laboratories. Age (OR 1.02 95% CI 1.01–1.02) and male sex (OR 1.26 95% CI 1.08–1.48) were predictors of a positive anti-HCV RDT test. Northern administrative regions in Ghana had the highest HCV seroprevalence ranging from 8.3–14.4%. Among 55, 458 potential blood donors, crude HCV seroprevalence was 3.57% (95% CI 3.42–3.72). Testing was through Rapid Diagnostic Test (RDT) kits in most facilities, and only 2 of 17 centres were performing HCV RNA testing. The median cost of an anti-HCV

laboratory and blood bank data. The relevant data are contained within the paper. Any other data are available from the Ghana Health Service and Teaching Hospitals included in this study for researchers who meet the criteria for access to confidential data.

**Funding:** Authors YAN, GG and YKA received partial funding. Funding agency: Coalition for Global Hepatitis Elimination at the Task Force for Global Health (https://www.globalhep.org/). The funders had no role in study design, data collection and analysis, decision to publish, or preparation of the manuscript.

**Competing interests:** The authors have declared that no competing interests exist.

RDT test was $0.97 (0–1.61) and $3.23 (1.61–7.58) for persons with and without government health insurance respectively. The median cost of a 12-week course of the pan-genotypic direct-acting antiviral therapy sofosbuvir-daclatasvir was $887.70. In conclusion, there are significant regional differences in HCV burden across Ghana. Limited access to and cost of HCV RNA and DAA therapy hinders testing and treatment capability, and consequently HCV elimination efforts. A national HCV program supported with a sustainable financing plan is required to accelerate HCV elimination in Ghana.

## Introduction

Each year, approximately 1.5 million new Hepatitis C virus (HCV) infections occur globally, and it is estimated that, as of early 2020, 56.8 million people in the world were living with chronic HCV infection [1, 2]. HCV infection remains a significant risk factor for the development of chronic and end-stage liver disease [3]. The World Health Organisation (WHO) estimates that in 2019, there were close to 300,000 HCV-related deaths, largely due to cirrhosis and primary hepatocellular carcinoma (HCC) worldwide [1]. Globally, the population attributable fraction (PAF) of HCV infection to HCC is approximately 20%, however, this differs depending on geographic location, with a PAF of 11% and 79% for eastern Asia and northern Africa respectively [4].

In sub-Saharan Africa (SSA), the estimated HCV viraemic prevalence is 0.8%, and about 9.2 million people in the region are living with HCV [5]. The burden of HCV varies by SSA region. Modelled estimates of viraemic HCV infection in southern Africa suggest a prevalence of 0.35–0.75%, whilst in western and several northern African countries, the viraemic prevalence is between 0.7–1.3% [2]. Notably within West Africa, modelled estimates suggest a higher burden of viraemic persons in Ghana and Burkina Faso, and the prevalence is reported to be between 1.3–2.3% [2]. There are few county-specific population-based studies on HCV within SSA and estimates of disease burden are limited by the accuracy of serological testing, and the limited availability of molecular tests to determine viraemia [6, 7].

Since the development of the global health sector strategy (GHSS) on viral hepatitis 2016–2021 [8], there has been a global effort to put in measures that will help countries reach elimination targets by the year 2030 [9]. Key strategies for achieving these goals include the development of national hepatitis control programs and the scaling up of HCV testing and treatment [10]. The GHSS 2022–2030 update includes targets for 90% of people living with HCV are diagnosed and that 80% are cured. Furthermore, an incidence target of 5 cases per 100,000 per year has been set [11], and although there has been a decline in global burden of HCV infection in recent years [2], it is unlikely that countries will achieve elimination targets by 2030 if more is not done to improve HCV testing and treatment at the country level [12]. The challenges of achieving HCV elimination targets in SSA include a lack of population-based screening programs, high cost and poor access to HCV RNA or core antigen testing, limited access to affordable treatment and lack of political will in some countries to address HCV care [7, 13].

In Ghana, the estimated national HCV seroprevalence in a systematic review of studies conducted between 1995 and 2015 was 3.0% [14], however more recent estimates of HCV burden are lacking. Furthermore, the existing national policy on viral hepatitis published in 2014 [15], is based on local data obtained before the year 2014. To determine progress towards HCV elimination and inform revised policies on HCV testing and treatment in Ghana, up-to-date

epidemiological data are required. Furthermore, an assessment of testing and treatment capacity, availability, and affordability are necessary to develop strategies that will address country-specific challenges which hinder the progress towards HCV elimination. This study therefore aimed to determine the HCV seroprevalence in Ghana, and the testing and treatment capacity related to HCV infection across public healthcare institutions in the country.

## Materials and methods

### Study design

A cross-sectional study was conducted to determine the HCV seroprevalence using laboratory and blood bank registers of public health institutions in Ghana. Secondly, a survey was conducted in which heads of laboratory and pharmacy services for each institution were asked to provide information on the types of HCV-related tests and medicines available, as well as the cost of tests and medicines in their facility for the year 2021.

### Sampling approach

Ghana Health Service hospitals, teaching hospitals, public health reference labs and faith-based institutions, providing HCV-related testing in each of Ghana's 16 administrative regions were eligible for data collection. The 16 administrative regions in the country were zoned into northern, middle, and southern zones, and a purposive sampling approach was used to select at least 2 two regional hospitals, 2 district hospitals, 1 faith-based hospital and 1 public health reference laboratory within each zone. Additionally all teaching hospitals (5 in total) were approached for data collection. Blood banks attached to each of the institutions formed the source of blood bank data.

### Data collection

Data collection took place between February 2021 and December 2021. During the data collection period, all laboratory and blood bank records for HCV-related testing performed between 1st January 2016 to 31st December 2021 in the various study sites were reviewed. In total, 103,609 laboratory register entries and 55,458 blood bank register entries were retrieved for this study. Registers from which data were obtained were paper based, with data either hand-written or typed electronically in Microsoft Word or Excel. Data were recorded in these registers as monthly or yearly aggregates, or alternatively, on a case-by-case basis for individual cases tested. Data abstracted for this study included total number of people tested for HCV, and the number of cases that were either positive or negative. Data for age, gender, and year of testing were also abstracted where available. Three field workers per study site had access to patient data, which contained patient name and hospital ID number, however no information that could identify individual participants was abstracted during or after data collection. Additionally at each institution, we conducted a survey on laboratory testing and treatment capacity, and cost of testing and treatment for HCV using a study questionnaire. The survey was conducted among the heads of the laboratory and pharmacy units of each institution. Data collected included price of HCV-related tests and medicines, types of testing available, average number of tests performed per month and limitations to testing.

### Statistical analysis

Descriptive statistics are reported including frequencies and percentages (categorical variables) and mean with standard deviation (continuous variables) are reported. Crude and pooled estimates are reported for HCV seroprevalence. Crude estimates were determined using total

persons positive for anti-HCV (numerator) divided by total persons tested (denominator) multiplied by 100. Pooled estimates were determined by treating each administrative region as a sub-group with inverse-variance weighting and recalculation of the overall prevalence [16]. Logistic regression was used to determine predictors of HCV seropositivity. For cost of testing and treatment, the median cost and interquartile range were determined. The frequency and proportion of tests and medicines available at each institution were also determined. To handle missing data, available case analysis was used. Data analysis was performed using Stata, version 17; StataCorp software.

## Ethics approval and consent to participate

No identifiable patient information was obtained, no patient enrolment for primary data was needed, and no informed consent from patients was required. Therefore, an exemption from requiring informed consent was granted by the Ghana Health Service Ethical Review Committee, Komfo-Anokye Teaching Hospital Institutional Review Board, Cape Coast Teaching Hospital Ethics Review Committee and Korle Bu Teaching Hospital Institutional Review Board. All methods were carried out in accordance with relevant guidelines and regulations. The study was approved by the ethical review committees of the Ghana Health Service (ERC number GHS-ERC 002/10/20), Korle Bu Teaching Hospital (KBTH-STC 00083/2021), Komfo-Anokye Teaching Hospital (KATH IRB/AP/099/21) and the Cape Coast Teaching Hospital (CCTHERC/EC/2021/117).

## Results

### Records reviewed and availability of data

Out of 23 sites selected for data collection, data was available for 17 (73.9%) sites (northern zone: 5 sites; middle zone: 6 sites; southern zone: 6 sites). Data was collected from 5 teaching hospitals, 6 regional hospitals, 3 district hospitals and 3 faith-based institutions, spanning 12 out of 16 administrative regions in Ghana. A total of 103,609 laboratory register entries from January 2016 to December 2021 were recorded. Of these, 22,876 entries were individual patient-level data whilst 80,733 were data reported from monthly or yearly aggregates. Additionally, 55,458 blood bank entries were reviewed from 13 blood banks. Data reported from each site varied by year because not all facilities could retrieve paper-based records for every year requested.

### HCV seroprevalence from hospital laboratory registers

Seroprevalence estimates were obtained from results from testing with rapid diagnostic test kits (RDT) recorded in laboratory registers. Out of 23,175 records in which gender information was available, males comprised 12,871 (55.5%) and females 10,304 (44.5%) of persons tested. Out of 19,752 records where age information was available, the median age of persons tested was 31 years (IQR 23–45). In total 2,721 out of 103,609 individuals were anti-HCV positive, representing a crude HCV seroprevalence of 2.62% (95% CI 2.53–2.72) across the country. The pooled HCV seroprevalence based on sample size per administrative region was 4.58% (95% CI 4.06–5.11). The HCV seroprevalence was lowest in the 0–11 years age group (2.13%) and highest in the 60+ years age group (7.59%) (Fig 1). Age was a predictor of a positive anti-HCV RDT test (OR 1.02 95% CI 1.01–1.02). Males were more likely to test positive than females (OR 1.26 95% CI 1.08–1.48) (Table 1).

When the burden across the country was evaluated, the highest HCV seroprevalence was found in the Northern zone i.e., in the Upper East (14.44%) and Upper West (13.54%) regions

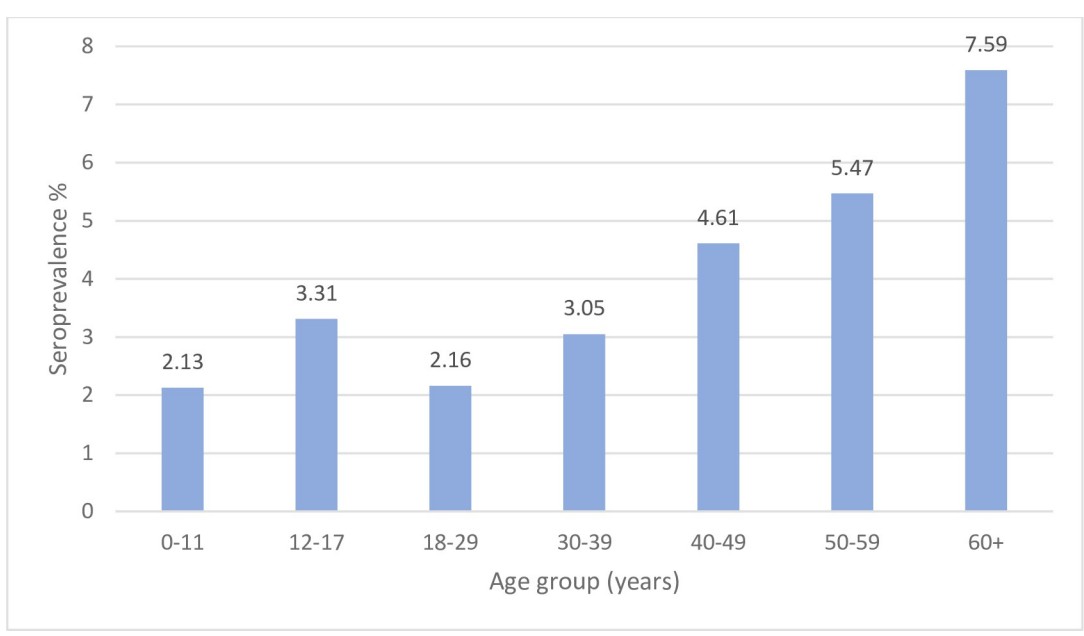

**Fig 1. Hepatitis C antibody (Anti-HCV) seroprevalence based on laboratory-based rapid diagnostic tests (RDTs) by age group, 2016–2021.** 0–11 years (n = 1126), 12–17 years (n = 906), 18–29 years (n = 7075), 30–39 years (n = 4132), 40–49 years (n = 2668) 50–59 years (n = 1736) 60+ years (n = 2109).

of Ghana (Fig 2). This trend was similar when the burden across regions was examined in children aged less than 18 years. Where data on age were available, it was found that seroprevalence was highest in the Upper East (7.8%) and Savannah (6.0%) regions, and lowest in the Southern Zone i.e., Greater Accra region (0.5%) (Fig 3).

**Table 1. Factors associated with HCV seropositivity among hospital attendants between 2016–2020.**

|  | Adjusted Odds Ratio* | 95% CI | P value |
|---|---|---|---|
| Sex |  |  |  |
| Male | 1.262 | 1.075–1.480 | 0.004 |
| Age (years) | 1.018 | 1.014–1.022 | <0.001 |
| Year |  |  |  |
| 2016 | Ref |  |  |
| 2017 | 0.39 | 0.19–0.80 | 0.01 |
| 2018 | 0.35 | 0.12–0.51 | <0.001 |
| 2019 | 0.44 | 0.22–0.88 | 0.02 |
| 2020 | 0.52 | 0.26–1.05 | 0.07 |
| Region |  |  |  |
| Volta | Ref |  |  |
| Central | 1.32 | 0.77–2.28 | 0.31 |
| Eastern | 1.02 | 0.58–1.79 | 0.95 |
| Greater Accra | 2.09 | 1.11–3.93 | 0.02 |
| Bono | 4.26 | 2.29–7.92 | <0.001 |
| Upper East | 11.45 | 6.76–19.40 | <0.001 |
| Western | 0.86 | 0.25–3.02 | 0.82 |
| Savannah | 5.48 | 2.87–10.46 | <0.001 |

*Multivariable model adjusted for age (continuous) gender (male, female), year (categorical) and region (categorical)

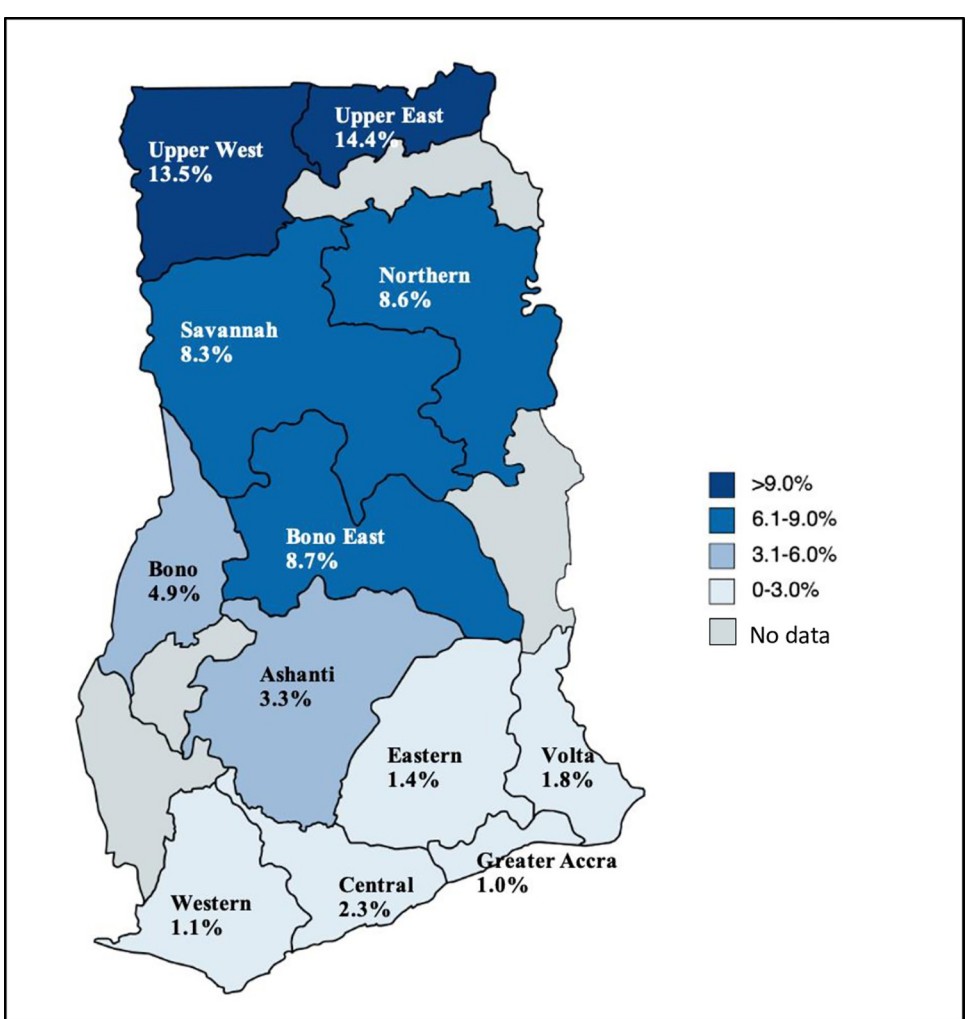

**Fig 2. HCV seroprevalence based on laboratory-based RDT tests by region, 2016–2021 (all ages).** Reprinted from [MapChart] under a CC BY license, with permission from mapchart.net, original copyright 2023.

### HCV seroprevalence from blood bank registers

A total of 1,980 out of 55,458 people screened at blood banks were anti-HCV positive, representing a crude seroprevalence of 3.57% (95% CI 3.42–3.72). The pooled HCV seroprevalence was 2.65% (95% CI 2.30–3.01). The regions with the highest seroprevalence were the Upper West region (11.28%) followed by the Upper East region (6.87%) (Fig 4).

### Laboratory capacity assessment

Out of 17 study sites, only two teaching hospitals were performing HCV RNA testing for establishing the presence of viremia (Table 2). Of these, one centre had capacity to perform the test on-site, and the other outsourced the testing to a private diagnostic company. In the remainder of the sites, patients were advised to use private diagnostic companies for HCV RNA testing. Only one facility conducted HCV genotyping. Rapid diagnostic testing was widely available across all sites; however, ELISA-based testing was limited to teaching (4/5) and a few regional hospitals (3/17). RDT kits available varied by brand, with some laboratory staff uncertain whether the test kits in use were approved by the Ghana Food and Drugs Authority. Examples

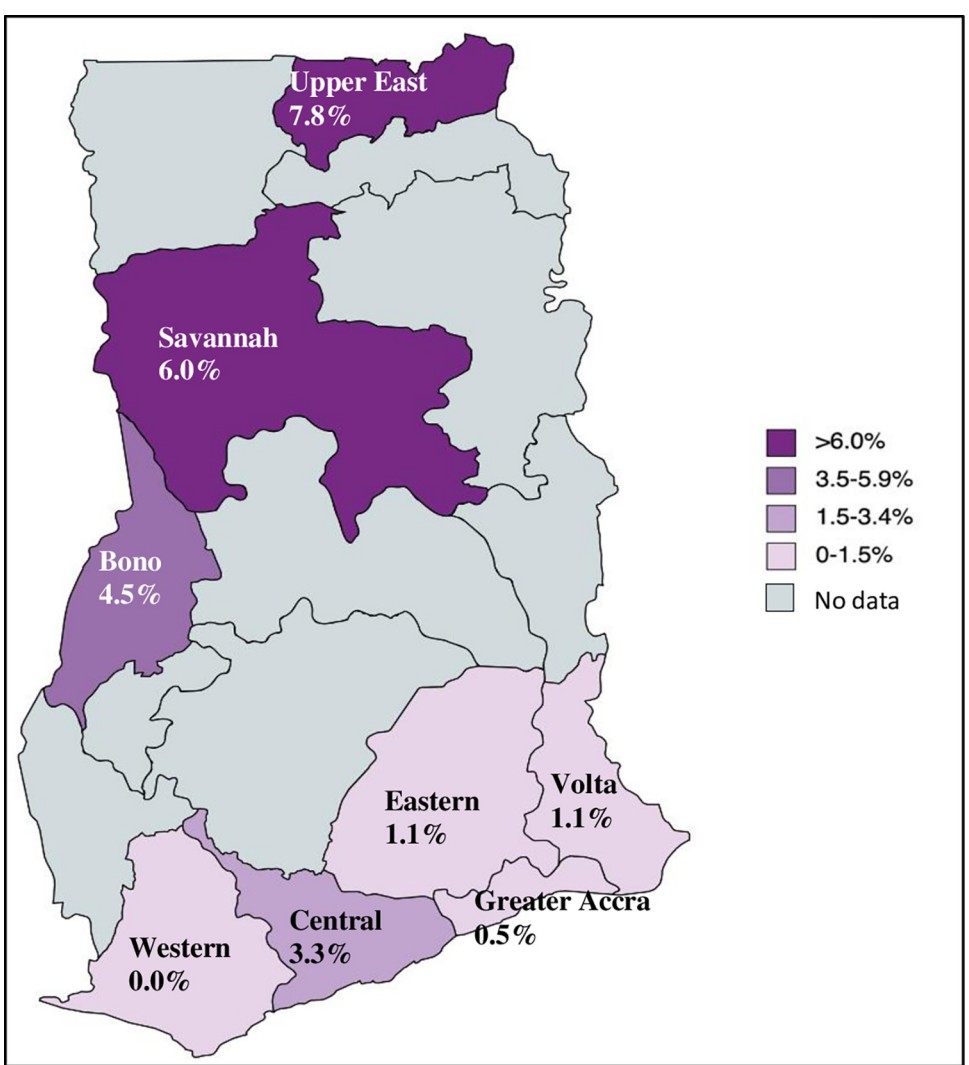

**Fig 3. HCV seroprevalence in children and adolescents (<18 years) based on laboratory-based RDT tests by region, 2016–2021.** Reprinted from [MapChart] under a CC BY license, with permission from mapchart.net, original copyright 2023.

of RDT brands used included DiaSpot, Wondfo, Abbott SD Bioline™, InTec, HighTop, and Global RDT. At the time of the study, the Ghana FDA list of accredited HCV RDT kits included only the Wondfo HCV kit. The Abbot SD Bioline and InTec RDT were listed in the WHO prequalified in vitro diagnostic products. No centre was performing HCV core antigen (cAg) testing. Direct-acting antiviral (DAA) therapy was not stocked in any hospital pharmacy visited; however, it was reported that these medicines could be obtained from private pharmacies if the patient could afford to pay out of pocket. Each hospital pharmacist asked referenced the same pharmacy/supplier located in Accra, the capital of Ghana, for purchase of sofosbuvir-daclatasvir.

## Cost of diagnosis and treatment

The price for anti-HCV testing was based on health insurance status. Cost of testing for both RDT and ELISA was subsidized if an individual had government health insurance; $0.97 vs

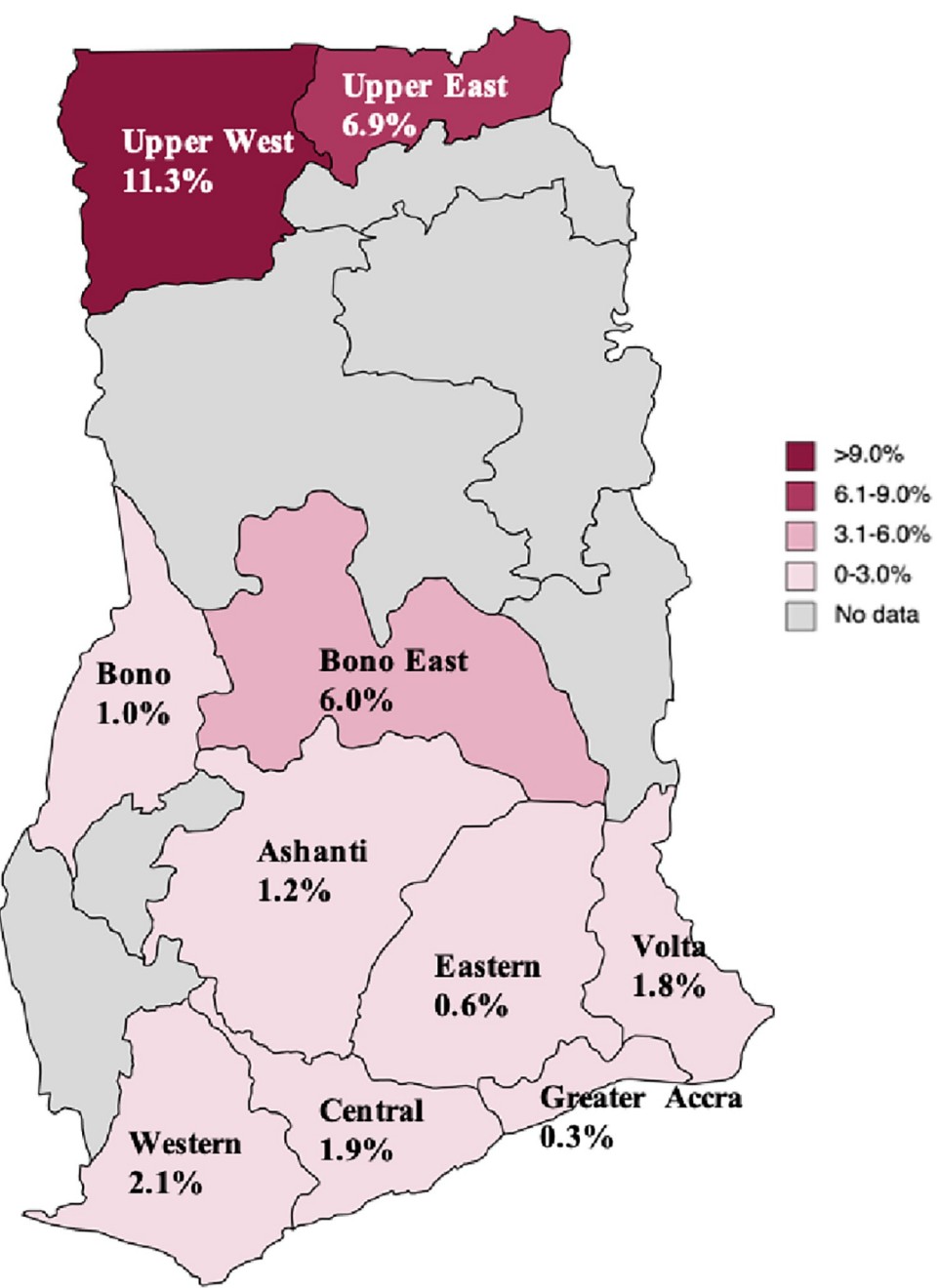

**Fig 4. HCV seroprevalence among potential blood donors by region based on RDT testing 2016–2020.** Reprinted from [MapChart] under a CC BY license, with permission from mapchart.net, original copyright 2023.

$3.23 for RDT and $3.15 vs $7.26 for ELISA for insured vs non-insured patients respectively. However, for HCV RNA, there was no subsidy on cost of testing, and patients had to pay out-of-pocket regardless of insurance status. It was reported by laboratory heads that the cost of testing was dependent on the price of the test from the supplier. DAA therapy was not subsidized by government health insurance. A 12-week course of pan-genotypic DAA sofosbuvir-daclatasvir was $887.70. Table 3 summarizes diagnostic and treatment costs.

**Table 2. Summary of HCV-related tests available by type of health facility (2021).**

|  | Teaching | Regional | District | CHAG | Total n/N (%) |
|---|---|---|---|---|---|
| *Diagnostics* |  |  |  |  |  |
| Anti-HCV (RDT) | 5/5 | 6/6 | 3/3 | 3/3 | 17/17 (100) |
| Anti-HCV (serology) | 4/5 | 3/6 | 0/3 | 0/3 | 7/17 (41.2) |
| HCV cAg | 0/5 | 0/6 | 0/3 | 0/3 | 0/17 (0) |
| HCV RNA* | 2/5 | 0/6 | 0/3 | 0/3 | 2/17 (11.8) |
| HCV Genotyping | 1/5 | 0/6 | 0/3 | 0/3 | 1/17 (5.9) |
| *Therapeutics* |  |  |  |  |  |
| Pan-genotypic therapy** | 0/5 | 0/6 | 0/3 | 0/3 | 0/17(0) |

HCV = Hepatitis C virus, RDT = Rapid Diagnostic Test, cAg = Core antigen, RNA = Ribonucleic acid.

*One site performed test on-site, and the other outsourced to a private laboratory.

**Medication was not available in the hospital pharmacy but was available on request from private pharmacies.

## Discussion

In this study we report a pooled HCV seroprevalence of 4.6% in patients tested at public health care facilities, which is slightly lower than reported in neighbouring countries. Seroprevalence estimates from a systematic review in Cameroon, in which 87% of studies included were facility-based, reported a pooled anti-HCV prevalence of 6.5% [17] whilst in north-east Nigeria, the reported seroprevalence among 560,857 out-patient clinic patients and 60,285 in-patient admissions attending a tertiary referral centre was 6.9%. In Ghana, the national HCV seroprevalence based on a systematic review by Agyeman et al in 2016 was 3.0% [14], however more recent nationwide estimates are not available. The difference in seroprevalence between this and Agyeman's study may be influenced by the type and quality of diagnostic tests used in studies included in the systematic review, including the sensitivity and specificity of test kits, or may reflect the different populations in the two studies. In the present study, data reviewed included that from hospital lab results of patients who may likely have been tested because of clinician suspicion of HCV infection or its related conditions such as chronic liver disease, or alternatively in the work-up for conditions in which HCV is highly co-morbid such as chronic renal failure or sickle cell disease. Notwithstanding these possibilities, it is important to note that the population tested in hospital laboratories also includes persons directed by clinicians

**Table 3. Summary of HCV testing and treatment costs.**

|  | Subsidized price with government health insurance USD Equivalent* Median (Range) | Non-subsidized price without government health insurance USD Equivalent* Median (Range) |
|---|---|---|
| *Diagnostics* |  |  |
| Anti-HCV (RDT) | 0.97 (0+ – 1.61) | 3.23 (1.61–7.58) |
| Anti-HCV (ELISA) | 3.15 (1.96–8.07) | 7.26 (5.65–50.03) |
| HCV RNA | Not subsidized | 88.77 (88.77–129.12) |
| *Therapeutic* |  |  |
| 12-week course of pan-genotypic therapy | Not subsidized | 887.70 |

*1 GHS = USD 0.1614 at the time of data collection

+Only one facility offered this test for free

to undergo testing for non-HCV-related conditions such as pregnant women, patients attending outpatient clinics with long term chronic conditions such as hypertension, and healthy individuals undergoing routine medical screening. On the other hand, the study by Agyeman and colleagues reviewed studies comprising significantly low-risk populations, with their estimate heavily influenced by large studies conducted among blood donors. This is likely to have led to a lower estimate of national seroprevalence in their study.

Upon comparison of the disaggregated HCV seroprevalence among blood donors in the Agyeman study of 2.6% to the anti-HCV prevalence among blood donors in this study, we found a similar estimate of 2.7%. Seroprevalence was likely lower among potential blood donors compared with the rest of the study population because prior to testing, individuals are routinely assessed for eligibility to donate using a standard screening form to eliminate persons who are likely to test positive for HIV, HBV, or HCV as per Ghana's national blood donation guidelines [18], thus making this population low-risk. Reported anti-HCV prevalence in studies from other countries in the SSA region among blood donors range from 0.8% in Ethiopia [19] to 2.32% in Mali [20], and 6.9% in neighbouring Burkina Faso [21].

This is the first study to explore the HCV seroprevalence in the majority of administrative regions in Ghana, since previous studies on HCV have largely been conducted in the Greater Accra and Ashanti regions [14, 22–26]. Significantly, there was unequal burden of disease across the different administrative regions, with regions in the northern zone demonstrating the highest seroprevalence (8.6–14.4%) and the Greater Accra Region in southern Ghana demonstrating lowest seroprevalence (1.0%). A similar pattern has been reported with the burden of Hepatitis B Virus (HBV) in Ghana [27]. Northern Ghana, compared with the rest of the country, has lower access to healthcare, weaker healthcare infrastructure, lower rates of hospital deliveries and lower doctor-to-patient and nurse-to-patient ratios [28–30], which may explain the higher burden of HCV in this part of the country. Furthermore, cultural practices such as scarification of the face and other parts of the body which may occur as early as the first week of life, for purposes of tribal and family identification, spiritual protection and traditional medicine use are more prevalent in the northern regions than in the south [31], and likely contribute to higher rates of HCV transmission in the region. In addition to this, chieftaincy and ethnic conflicts which occur at a higher rate in Northern Ghana, may lead to increased HCV burden directly through blood exposure or indirectly through weakened socioeconomic and health infrastructure [32, 33]. Southern Ghana was found to have the lowest anti-HCV prevalence in this study. In the Greater Accra region, a 2020 study conducted in a public hospital among 728 patients reported an HCV seroprevalence of 1.6% [26], close to our reported estimate in the same region of 1.0%.

Several factors may contribute to the HCV burden in Ghana. This includes a poor level of HCV knowledge and awareness in the population, which may mean individuals are less likely to be aware of their HCV status and therefore would be less likely to undertake practices to limit spread [6]. For example, studies in the Ashanti region demonstrated that the majority of study participants had never heard of HCV and were unaware of its modes of transmission [24, 34]. It may be possible that healthcare provider knowledge on HCV is also inadequate, since some studies suggest knowledge gaps in HBV-related care among providers in Ghana [35–37], however specific studies on knowledge of HCV among healthcare workers in Ghana are lacking. Other factors such as blood to blood exposure including through scarification as previously described, unsafe male circumcision practices and intravenous drug use (IVDU) are potential contributory factors to HCV infection in Ghana and remain a probable mode of transmission [38]. The degree to which IVDU is prevalent in Ghana is not well established, however a study among inmates reported that roughly one third of inmates had a history of intravenous drug use [39]. In a study of 323 person who inject drugs (PWID) and persons who

use drugs (PWUD) conducted in four regions in Ghana, HCV seroprevalence was reported to be 5.6% [40]. There is evidence to suggest that a significant proportion of PWID in Ghana reuse and share needles due to the high cost and difficult access [39, 41]. Currently there are limited harm reduction programmes for PWID in Ghana, with no formal syringe exchange programmes for this key population [38].

A further mode of HCV spread may be through vertical transmission from pregnant women to their babies. Studies among pregnant women have demonstrated seroprevalence data ranging from 2.7% in the Central region [42] to 7.7% in a study in the Ashanti region [23]. Although anti-HCV testing is recommended as part of routine antenatal care screening in Ghana, this test is not free in many public health facilities, and a proportion of pregnant women may not be able to pay out-of-pocket. For example, in this study, only one centre offered the anti-HCV test at no charge for insured patients. Furthermore, unlike Human Immunodeficiency Virus (HIV), HBV, and syphilis, neither the maternal health record book nor the labour ward registers in Ghana require recording of HCV status, therefore midwives or antenatal clinic nurses may overlook HCV testing during pregnancy and delivery.

This study found a seroprevalence of 2.13% among children aged 0–11 years and 3.31% among those aged 12–17 years attending healthcare facility laboratories. A recent global systematic review reported an anti-HCV seroprevalence in African children (<20 years old) of 3.02%, with seropositivity of 2.45% in those aged <10 years and 4.74% in those between ages 10–20 years [43]. In Ghana, a study at the Princess Marie Louis Children's Hospital in the Greater Accra Region reported a seroprevalence of 0.5% from a hospital population of 200 children, comparable to this study's finding of 0.5% in the same region [25]. The anti-HCV prevalence found in this study, particularly in the Bono, Savannah and Upper East regions demonstrate the need to include eligible children (above 3 years of age) and adolescents in a screening and treatment program for HCV in Ghana, in line with current guidance [44, 45].

Although RDT kits were widely available in all sites visited, it was concerning to note that laboratory personnel were uncertain whether these kits were either approved for use by the Ghana Food and Drugs Authority or WHO pre-qualified. A previous study found that out of 17 different HCV RDT kits used in 374 public and private diagnostic laboratories in Ghana, only 2 (11.8%) were WHO pre-qualified [46]. At study sites visited, procurement of test kits was handled at the facility level by procurement officers or laboratory personnel, with no direct input from the Ghana National Viral Hepatitis Control Program nor Ministry of Health. It is known that the performance of RDT kits is variable, and sensitivity may range from 75% to 100% [47, 48]. Consequently, the use of non-approved RDT kits may increase the chances of false negative or false positive anti-HCV results, which may therefore under- or overestimate HCV seropositivity if this method is used as the sole screening tool [49]. The variability in price of anti-HCV kits was also of concern, and this is likely due to the different brands of kits used, since laboratory personnel reported that pricing was dependent on the price from the supplier. A specific policy on test kit procurement involving purchase and subsequent distribution by the Ghana National Viral Hepatitis Control Program will not only ensure that FDA approved or WHO-prequalified kits are used but may also bring some stability to pricing in public health facilities. Furthermore, the poor availability of ELISA testing in sites visited emphasizes the need to ensure the use of pre-qualified HCV RDTs.

PCR testing for HCV RNA was only available in two centres visited, with one of these outsourcing to a private laboratory. Qualitative or quantitative HCV RNA testing is crucial for determining which patients require direct-acting antiviral (DAA) therapy. In the absence of PCR capacity in public hospitals, patients must often patronise private laboratories. In addition to limited availability, the high cost of $88.7 found in this study poses a significant barrier to HCV treatment in Ghana [6, 14, 46]. If Ghana is to achieve scale-up of testing and treatment,

there is a pressing need to increase PCR testing capacity. One way may be to leverage the improved PCR testing capacity in some public health facilities in response to the COVID-19 pandemic. Furthermore, there is a need to decentralise testing to, at a minimum, regional hospital level. To increase testing access, it may also be necessary to consider alternate methods for testing, including the use of dried blood spot sampling (DBS), which is cheaper and less vulnerable to strict cold-chain storage and transfer requirements [50], in place of venous blood sampling. Furthermore, development of testing algorithms based on HCV core antigen testing [51], which is currently not available in Ghana, may be an alternate way to scale-up testing and treatment in the country.

In this study, no hospital visited had stock of DAA medication in their pharmacies, but it was noted that these drugs could be obtained from privately run pharmacies if the patient could afford the treatment. There are currently no government subsidies on the cost of medications and current pricing in the country for a 12-week course of pan-genotypic therapy appears higher than in other African countries such as Nigeria and Cameroon ($750) [52]. A 2022 cost-utility analysis in four African countries estimated the generic price for a 12-week course of sofosbuvir/daclatasvir to be $195 and for sofosbuvir/velpatasvir to be $450 [53]. For many patients requiring HCV treatment in Ghana, personal income may be insufficient to cover the current costs of diagnosis and treatment. Relying on complete government financing may also not be practical or sustainable [52]. To improve treatment access, strategies to overcome these costs are necessary, and may include shared financing between governments and individuals, improved global access programs, reduced pricing by large diagnostic and pharmaceutical companies and increased advocacy by civil society groups and patients to expand access to care. In response to the need for provision of free or affordable DAA therapy to infected persons to enable HCV elimination, the Ghana Health Service, through a donation from the Ministry of Health and Population of Egypt, has embarked on a free treatment project entitled 'Screening and Treatment Opportunity Project for Hepatitis C (STOP Hep C) in Ghana. The project, which was officially launched in March 2023, aims to provide a free 12-week course of Sofosbuvir/ Daclatasvir to 50,000 patients with viraemic HCV infection [54].

The strengths of this study include the broad coverage of administrative regions in Ghana, and the inclusion of data different types and levels of public health facilities, which provided previously unreported data on HCV seroprevalence in certain regions in the country. Furthermore, we were able to provide age-related estimates of HCV seropositivity, highlighting the need to include children in any HCV screening and treatment program in Ghana. Another major strength of this study is that we were able to collect information on capacity and pricing of HCV testing and treatment in Ghana, which can directly inform policy in the country.

Limitations of this study included the use of secondary data, which meant that not all institutions were able to provide data for all the years of interest. In addition to this, the majority of the data was aggregated, which limited the ability to assess for risk factors in the study population. Furthermore, the study population comprised hospital attendants, whose seroprevalence estimates may be higher than that of the general population. There is also a need to assess the prevalence of viraemic HCV infection, rather than seroprevalence at the national level since such population-based data are lacking in Ghana. Additionally, the use of seroprevalence data may overestimate the HCV prevalence, since anti-HCV positivity comprises those with active infection, resolved infection or a false positive test result. Finally, the use of varied brands of RDT test kits with different sensitivity and specificity may have affected the accuracy of prevalence estimates.

## Conclusions

The uncertainty of the true national HCV prevalence in the general Ghanaian population emphasizes the need for additional population-based studies to improve disease burden estimation, including the HCV incidence, viraemia prevalence, and mortality associated with HCV in the country. Possible solutions will be to undertake strategies such as testing of stored population-based samples or undertaking a national testing campaign for HCV. In this study, we have identified that there are significant regional differences in HCV burden across Ghana, with the northern regions demonstrating the highest HCV seroprevalence. Targeting prevention, testing and treatment policies for northern Ghana may therefore be warranted. Limited access to and cost of HCV RNA testing and DAA therapy hinders testing and treatment capability, and consequently HCV elimination efforts. There may be a need to improve HCV awareness in Ghana, through multiple avenues including a national campaign by the Ministry of Health. An improved policy on RDT kits is required, with measures put in place to ensure that only Ghana's FDA or WHO prequalified test kits are used in the public and private sectors. Finally, a national program for HCV elimination including a financing plan for sustainability is important if Ghana is to achieve the 2030 viral hepatitis elimination targets.

## Acknowledgments

The authors would like to acknowledge the following people and groups:

- Hepatitis Foundation of Ghana

- Hepatitis Society of Ghana

- Ghana Association for the Study of Liver and Digestive Diseases

- Ghana Field Epidemiology and Laboratory Training Program, School of Public Health, University of Ghana

- All individuals who contributed data to the HEAT Project and participated in the project launch and stakeholder's meetings

## Author Contributions

**Conceptualization:** Yvonne Ayerki Nartey, Atsu Seake-Kwawu, Kafui Senya, Amoako Duah, John W. Ward, Dorcas Obiri-Yeboah.

**Data curation:** Yvonne Ayerki Nartey, Rafiq Okine, Atsu Seake-Kwawu, Georgia Ghartey, Yaw Karikari Asamoah, Ampem Darko Jnr Siaw, Amoako Duah, Alex Owusu-Ofori, Opei Adarkwa, Seth Agyeman, Sally Afua Bampoh.

**Formal analysis:** Yvonne Ayerki Nartey, Rafiq Okine.

**Funding acquisition:** Yvonne Ayerki Nartey, Atsu Seake-Kwawu, Lindsey Hiebert, John W. Ward, Lewis Rowland Roberts.

**Investigation:** Yvonne Ayerki Nartey, Georgia Ghartey, Yaw Karikari Asamoah, Kafui Senya, Amoako Duah, Alex Owusu-Ofori, Opei Adarkwa, Lindsey Hiebert, Ansumana Sandy Bockarie, Yaw Asante Awuku, Dorcas Obiri-Yeboah.

**Methodology:** Yvonne Ayerki Nartey, Rafiq Okine, Atsu Seake-Kwawu, Georgia Ghartey, Yaw Karikari Asamoah, Ampem Darko Jnr Siaw, Kafui Senya, Amoako Duah, Alex Owusu-Ofori, Opei Adarkwa, Seth Agyeman, Sally Afua Bampoh, Lindsey Hiebert, John

W. Ward, Lewis Rowland Roberts, Ansumana Sandy Bockarie, Yaw Asante Awuku, Dorcas Obiri-Yeboah.

**Project administration:** Yvonne Ayerki Nartey, Atsu Seake-Kwawu, Kafui Senya, Alex Owusu-Ofori, Opei Adarkwa, Lindsey Hiebert, Neil Gupta, John W. Ward.

**Resources:** Lindsey Hiebert, Henry Njuguna, Neil Gupta, John W. Ward, Lewis Rowland Roberts.

**Software:** Lindsey Hiebert, Henry Njuguna, Neil Gupta, John W. Ward.

**Supervision:** Atsu Seake-Kwawu, Kafui Senya, Amoako Duah, Sally Afua Bampoh, Lindsey Hiebert, Henry Njuguna, Neil Gupta, John W. Ward, Lewis Rowland Roberts, Ansumana Sandy Bockarie, Yaw Asante Awuku, Dorcas Obiri-Yeboah.

**Validation:** Rafiq Okine, Georgia Ghartey, Yaw Karikari Asamoah, Ampem Darko Jnr Siaw, Alex Owusu-Ofori, Seth Agyeman.

**Writing – original draft:** Yvonne Ayerki Nartey.

**Writing – review & editing:** Yvonne Ayerki Nartey, Rafiq Okine, Atsu Seake-Kwawu, Georgia Ghartey, Yaw Karikari Asamoah, Ampem Darko Jnr Siaw, Kafui Senya, Amoako Duah, Alex Owusu-Ofori, Opei Adarkwa, Seth Agyeman, Sally Afua Bampoh, Lindsey Hiebert, Henry Njuguna, Neil Gupta, John W. Ward, Lewis Rowland Roberts, Ansumana Sandy Bockarie, Yaw Asante Awuku, Dorcas Obiri-Yeboah.

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
