## [Decision Letter · Decision Letter 0]

5 May 2023

PONE-D-23-08352Hepatitis C virus seroprevalence, testing, and treatment capacity in public health facilities in Ghana, 2016 – 2021; A multi-centre cross-sectional study.PLOS ONE

Dear Dr. Nartey,

Thank you for submitting your manuscript to PLOS ONE. After careful consideration, we feel that it has merit but does not fully meet PLOS ONE’s publication criteria as it currently stands. Therefore, we invite you to submit a revised version of the manuscript that addresses the points raised during the review process.

We look forward to receiving your revised manuscript.

Kind regards,

Olatunji Matthew Kolawole, Ph.D.

Academic Editor

PLOS ONE

Journal Requirements:

4. We note that Figures 2,3 and 4 in your submission contain [map/satellite] images which may be copyrighted. All PLOS content is published under the Creative Commons Attribution License (CC BY 4.0), which means that the manuscript, images, and Supporting Information files will be freely available online, and any third party is permitted to access, download, copy, distribute, and use these materials in any way, even commercially, with proper attribution. For these reasons, we cannot publish previously copyrighted maps or satellite images created using proprietary data, such as Google software (Google Maps, Street View, and Earth). For more information, see our copyright guidelines: http://journals.plos.org/plosone/s/licenses-and-copyright.

a. You may seek permission from the original copyright holder of Figures 2,3 and 4 to publish the content specifically under the CC BY 4.0 license.  

Additional Editor Comments:

Attend to the queries made by Review 1 and kindly provide succinct responses to the comments.

Reviewers' comments:

Reviewer's Responses to Questions

**Comments to the Author**

1. Is the manuscript technically sound, and do the data support the conclusions?

Reviewer #1: Yes

Reviewer #2: Yes

2. Has the statistical analysis been performed appropriately and rigorously? 

Reviewer #1: Yes

Reviewer #2: Yes

3. Have the authors made all data underlying the findings in their manuscript fully available?

Reviewer #1: Yes

Reviewer #2: Yes

4. Is the manuscript presented in an intelligible fashion and written in standard English?

Reviewer #1: Yes

Reviewer #2: Yes

5. Review Comments to the Author

Reviewer #1: 1. The rapid test kits used in this sero-prevalence study of hepatitis C need to have an evidence of post market verification using a serological test gold standard such enzyme immuno assay or multiplex beads assay as a gold standard.

2. Are remnants of the donors samples archived in these blood banks in order to provide answers to some of these questions? If yes, is recommended that you compare the serologically positive samples with the nucleic acid testing or at least EIA results. The sensitivity, specificity, positive predictive vale and the negative predictive value from the evaluation/ post market verification of these rapid test kits are quite essential to the reliability of this sero-prevalence data.

3. Does the record in the blood banks indicate the route of transmission of hepatitis C among the positive donors? This will help in prevention of Hepatitis C infection in Ghana.

Reviewer #2: As a Medical Laboratory Scientist , I enjoyed every bit of the article. The research methodology and data analysis was brilliant. But I beg different will that 17% data available would not be able to give a good representation of the prevalence of HCV infection in Northern Ghana. It is a brilliant idea, I hope this findings from this research is submitted to relevant stakeholders to help formulate policies that build a good disease surveillance for JCV infection in Ghana

6. PLOS authors have the option to publish the peer review history of their article (what does this mean?). If published, this will include your full peer review and any attached files.

Reviewer #1: **Yes: **Ado Garba Abubakar

Reviewer #2: No

---

## [Author Response · Author response to Decision Letter 0]

24 May 2023

Reviewer #1: 

1. The rapid test kits used in this sero-prevalence study of hepatitis C need to have an evidence of post market verification using a serological test gold standard such enzyme immuno assay or multiplex beads assay as a gold standard.

a. We agree with the reviewer. There was variability in test kits used by study sites over the testing period studied, therefore we could not obtain information on test kit performance/verification. The pitfalls of failure to use WHO prequalified kits is discussed on page 17 line 358-360 and in the limitations on page 19 line 410-412

2. Are remnants of the donors samples archived in these blood banks in order to provide answers to some of these questions? If yes, is recommended that you compare the serologically positive samples with the nucleic acid testing or at least EIA results. The sensitivity, specificity, positive predictive vale and the negative predictive value from the evaluation/ post market verification of these rapid test kits are quite essential to the reliability of this sero-prevalence data.

a. Thank you for your suggestion. This will be a good next step for this study; therefore, a new proposal will be written for ethical approval to undertake such a study. For the current paper, we are unfortunately unable to perform this analysis as we do not have permission to obtain any archived samples from 2016-2021.

3. Does the record in the blood banks indicate the route of transmission of hepatitis C among the positive donors? This will help in prevention of Hepatitis C infection in Ghana.

a. The records in the blood-bank made available to us did not contain routes of transmission among positive donors. We are currently conducting a study to investigate the risk factors associated with HCV seropositivity in Ghana.

Reviewer #2: 

1. As a Medical Laboratory Scientist , I enjoyed every bit of the article. The research methodology and data analysis was brilliant. But I beg different will that 17% data available would not be able to give a good representation of the prevalence of HCV infection in Northern Ghana. It is a brilliant idea, I hope this findings from this research is submitted to relevant stakeholders to help formulate policies that build a good disease surveillance for JCV infection in Ghana

a. Thank you for your review and contribution. 

Reviewer #3: 

1. It will be of great benefit to the manuscript/audience if the authors could provide information on any HCV-related intervention or policies the Ghana health services have introduced during the study period. With an elimination target in view, I believe every country would have initiated some action/intervention. The data presented suggest an increase in HCV infections – this could result from no intervention being conducted or interventions not doing what they are intended to. We can only arrive at such a conclusion if we have an idea of what interventions have been put in place.

a. Thank you for your suggestion. The authors are not aware of a specific HCV intervention undertaken during the study period. Information regarding a 2023 HCV-specific intervention by the Ghana Health Service has been included on page 18 lines 389-394. 

2. To estimate prevalence, the authors rely on serological data. However, the authors also mentioned access to molecular data although on a small scale at 1-2 health facilities. Was this molecular data included in estimating the seroprevalence or most of these individuals with molecular test results were initially tested with a serological test? The authors should provide information on how the molecular test results were processed.

a. We stated that “PCR testing for HCV RNA was only available in two centres visited, with one of these outsourcing to a private laboratory.” In both centers, this test had only become available a few months before the data collection period and even then, availability was dependent on whether there were cartridges to run the sample. There were only a handful of results available, which were too few to use for data analysis (fewer than 10 at the time of data collection). We therefore opted to use only seroprevalence data for this study. For individuals with molecular tests, it could not be ascertained whether the serological test was conducted within the same facility or upon request from a referral facility. 

3. The study authors need to revise and standardize some minor aspects of the manuscript.

For example, Materials and Method section (abstract) On page 5 line 108 the authors mentioned that the study was conducted in public health institutions (that could be interpreted as government-run facilities) but on page 6 lines 113 – 118, the private and faith-based health facilities were included in the review.

a. The following institutions were stated: “Ghana Health Service hospitals, teaching hospitals, public health reference labs and faith-based institutions”. These facilities visited all fell under the public health space and not the private sector. No private institution was visited for data collection. 

Furthermore, were the 16 admin regions grouped into 3 zones (northern, middle, and southern) from which the health facilities were selected. If yes, could the authors rephrase this section to better highlight what was done? Could the authors also provide information on the final number of facilities reviewed per zone? 

a. We have rephrased the section as suggested (page 6 lines 115-120). The information on final number per zone has been updated in the results section on page 8 lines 164-165

In the Method section (~ 6 per zone = 18 but in the results section, the authors mentioned 23 health facilities. What of the blood banks)

a. A purposive sampling approach was used to select at least 2 two regional hospitals, 2 district hospitals, 1 faith-based hospital and 1 public health reference laboratory, - this makes up 6 per zone totaling 18 sites. Additionally, we gathered data from, “...all teaching hospitals located within each zone”. There are 5 teaching hospitals in the country, making up 23 health facilities as stated in the results. Out of the 23 sites approached, laboratory data was available for 17 institutions and 13 blood banks. The section has been updated for clarity (page 6 lines 115-120)

Interestingly, in the result sections the zones are not mentioned. So why did the authors adopt a zoning approach that was eventually not used? I will suggest the zoning aspect be deleted. It is of no significance.

a. Zoning was purposively done to ensure that a range of public health facilities in each part of the country were included in data collection in order to have a dataset that spanned all areas of Ghana. 

b. Additionally, zoning helped to illustrate that the administrative regions in northern Ghana appeared to have a higher seroprevalence compared to the other two zones. Wording has been updated to reflect this on page 9, lines 192, 196, 287, 289. The discussion has previously discussed potential reasons for the difference in seroprevalence in Northern Ghana on page 14 lines 290-300.

4. As a study limitation, does the use of seroprevalence data overestimate the prevalence of HCV? Lines 73-78 as indicate a lower disease burden when viraemic data is considered. Seroprevalence data could reflect three possible scenarios 1) active infection 2) a resolved past infection and 3) a false positive. I will suggest the authors highlight this shortcoming in their analysis.

a. This is well noted. The following has been added to the study limitations on page 19 lines 408-410 “Additionally, the use of seroprevalence data may overestimate the HCV prevalence, since anti-HCV positivity comprises those with active infection, resolved infection or a false positive test result.”

5. What is the added value of figure 3 that highlights cases < 18 years of age? The data presented by the authors suggest a higher burden as age increases. If a more granular analysis was required, the focus should have been on the elderly group. What do the authors wish to highlight in the <18 group that is more important for this group?

a. We wish to highlight the HCV prevalence in the under 18 age-group since the WHO treatment guidelines were recently updated to include guidance on treating children with HCV infection.

6. Minor language corrections

Introduction Line 78 page 4: modify “high burden of viremia.” to “high burden of viraemic persons” ; 

Introduction Line 91 page 5: modify “The challenges to HCV response” to “The challenges of achieving the HCV elimination targets” 

Methods section: Line 117 page 13 correct “refence” to “reference” laboratory

Include a reference for the Agyeman and colleagues study mentioned in line 255.

All the suggested modifications and corrections have been effected, thank you.

7. Discussion section: The statement of HCV knowledge among healthcare workers is at best speculative. If not supported by a reference. The statement should either be referenced or deleted if no reference is available. Lines 296-298 

a. The following sentence has been deleted: 

“If the same is true, and HCV knowledge among healthcare workers in Ghana is insufficient, this may mean that healthcare workers may be less likely to screen, treat, or link HCV patients to appropriate care.”

---

## [Editor Report · Decision Letter 1]

8 Jun 2023

Hepatitis C virus seroprevalence, testing, and treatment capacity in public health facilities in Ghana, 2016 – 2021; A multi-centre cross-sectional study.

PONE-D-23-08352R1

Dear Dr. Nartey,

We’re pleased to inform you that your manuscript has been judged scientifically suitable for publication and will be formally accepted for publication once it meets all outstanding technical requirements.

Kind regards,

Olatunji Matthew Kolawole, Ph.D.

Academic Editor

PLOS ONE
---

## [Editor Report · Acceptance letter]

15 Jun 2023

PONE-D-23-08352R1 

Hepatitis C virus seroprevalence, testing, and treatment capacity in public health facilities in Ghana, 2016 – 2021; A multi-centre cross-sectional study. 

Dear Dr. Nartey:

I'm pleased to inform you that your manuscript has been deemed suitable for publication in PLOS ONE. Congratulations! Your manuscript is now with our production department. 

Kind regards, 

on behalf of

Dr. Olatunji Matthew Kolawole 

Academic Editor

PLOS ONE